# GOLD: A Global and Local-aware Denoising Framework for Commonsense Knowledge Graph Noise Detection

**Zheye Deng, Weiqi Wang, Zhaowei Wang, Xin Liu, Yangqiu Song**

Department of Computer Science and Engineering, HKUST, Hong Kong SAR, China

{zdengah, wwangbw, zwanggy, xliucr, yqsong}@cse.ust.hk

## Abstract

Commonsense Knowledge Graphs (CSKGs) are crucial for commonsense reasoning, yet constructing them through human annotations can be costly. As a result, various automatic methods have been proposed to construct CSKG with larger semantic coverage. However, these unsupervised approaches introduce spurious noise that can lower the quality of the resulting CSKG, which cannot be tackled easily by existing denoising algorithms due to the unique characteristics of nodes and structures in CSKGs. To address this issue, we propose GOLD (**Glo**bal and **L**ocal-aware **D**enoising), a denoising framework for CSKGs that incorporates entity semantic information, global rules, and local structural information from the CSKG. Experiment results demonstrate that GOLD outperforms all baseline methods in noise detection tasks on synthetic noisy CSKG benchmarks. Furthermore, we show that denoising a real-world CSKG is effective and even benefits the downstream zero-shot commonsense question-answering task. Our code and data are publicly available at https://github.com/HKUST-KnowComp/GOLD.

## 1 Introduction

The emergence of Commonsense Knowledge Graphs (CSKGs) has significantly impacted the field of commonsense reasoning (Liu et al., 2021; Zhang et al., 2020) as CSKGs provide commonsense knowledge that is often not explicitly stated in the text and difficult for machines to capture systematically (Davis and Marcus, 2015). While existing methods bank on expensive and time-consuming crowdsourcing to collect commonsense knowledge (Sap et al., 2019a; Mostafazadeh et al., 2020), it remains infeasible to obtain CSKGs that are large enough to cover numerous entities and situations in the world (He et al., 2022; Tandon et al., 2014). To overcome this limitation, various automatic CSKG construction methods have been

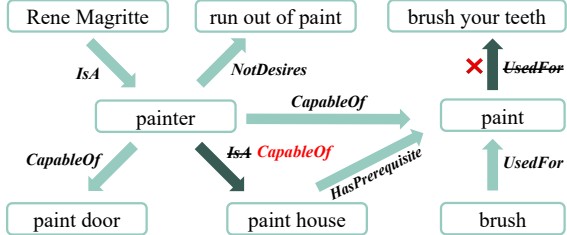

Figure 1: A subgraph of CSKG with two instances of noise. One noise is (*paint*, UsedFor, *brush your teeth*), as there should be no relation between these two nodes. Another noise is (*painter*, IsA, *paint house*) because the correct relation should be CapableOf.

proposed to acquire commonsense knowledge at scale (Bosselut et al., 2019), including prompting Large Language Model (LLM) (West et al., 2022; Yu et al., 2022), rule mining from massive corpora (Tandon et al., 2017; Zhang et al., 2022a), and knowledge graph population (Fang et al., 2021a,b, 2023). Although those methods are effective, they still suffer from noises introduced by construction bias and the lack of human supervision. Therefore, how to identify noise in large-scale CSKG accurately and efficiently becomes a crucial research question.

To tackle this issue, noise detection algorithms have been proposed for conventional entity-based KGs by primarily adopting two approaches: learning-based and rule-based. Learning-based methods like TransE (Bordes et al., 2013) learn representations of entities and relations that adhere to specific relation compositions like translation assumption or relational rotation. To enhance their performance, researchers also incorporate local information around the head and tail entities, such as different paths from head to tail (Lin et al., 2015; Xie et al., 2018; Jia et al., 2019) and neighboring triples (Zhang et al., 2022b). These methods aim to improve their ability to capture the complex relationships between entities in KGs. However,

they are not easily adaptable to the unique characteristics of CSKGs. In CSKGs, nodes are non-canonicalized, free-form text, meaning nodes with different descriptions may have related semantics. As illustrated in Figure 1, "*paint door*" and "*paint house*" are two distinct nodes but imply related semantics (Speer et al., 2017). Additionally, when detecting noise (*paint*, UsedFor, *brush your teeth*), "*brush your teeth*" is an isolated node that cannot be distinguished based on any structural information. Only through the power of a language model can it be learned that "*paint*" and "*brush your teeth*" are uncorrelated, thus detecting such noise. The aforementioned methods overlook this semantic information and cannot generalize to semantically similar events with diverse structural information.

On the other hand, rule-based methods utilize logical rules in KGs for noise detection. For instance, as shown in Figure 1, the correct relation between "*painter*" and "*paint house*" should be CapableOf. This can be easily detected through the learned logical rule: $\text{CapableOf}(x, y) \leftarrow \text{CapableOf}(x, z) \wedge \text{HasPrerequisite}(y, z)$. Belth et al. (2020) similarly propose an approach based on information theory that extracts sub-graph patterns to identify the noise. However, the sparsity of edges in CSKGs (Malaviya et al., 2020) posits a serious challenge to learning structural information well, as the number of learnable rules decreases significantly. This requires a generalizable rule-learning ability at the noise detector side to expand the rule bank accordingly, which is currently lacking. Therefore, applying noise detection models for KGs directly to CSKGs can result in incomplete learning of both semantic and structural information in the CSKGs.

In order to detect noises in CSKGs effectively, it is important to consider both the semantic information and the global and local structural information jointly. However, these factors have not been given enough importance in existing denoising approaches. To address this gap, we propose GOLD (**Glo**bal and **L**ocal-aware **D**enoising), a CSKG noise detector that uses a PLM-based triple encoder and two noise detectors that take into account both global and local structures (Section 4). Specifically, the triple encoder extracts the semantic information contained in the free-text formatted nodes in CSKGs. To identify correct patterns, the global detector uses high-frequency patterns extracted through rule mining, which intrinsically uses a

rule encoder to generalize the learned rules and guide noise detection. The local detector, inspired by Zhang et al. (2022b), adopts a graph neural network to efficiently measure the similarity of aggregated semantic information of neighboring triples of the head and tail nodes to help detect noise. Extensive experiments on two manually synthesized noisy-CSKG benchmarks demonstrate the efficacy and state-of-the-art performance of GOLD. Further experiments and analyses with ATOMIC[10x] (West et al., 2022), a large-scale CSKG distilled from GPT3, demonstrates its proficiency in identifying noise within real-world CSKGs, while also yielding advantages in the downstream zero-shot commonsense question-answering task.

In summary, in this paper, we make the following contributions:

- We introduce a new task: CSKG denoising, which can be applied to various CSKG construction and LLM distillation works.
- We propose a novel framework GOLD, which outperforms all existing methods (Section 6.1) and LLMs (Section 6.3).
- We show that GOLD successfully detects noises in real-world CSKGs (Section 6.5) and such denoising extrinsically benefits downstream zero-shot commonsense question-answering task (Section 6.4).

## 2 Related Work

### 2.1 Knowledge Graph Noise Detection

Many existing knowledge graph noise detection approaches utilize some local information while simultaneously training embeddings to satisfy the relational assumption. Path information is the most commonly used type of local information, as the reachable path from the head entity to the tail entity has been proven crucial for noise detection in knowledge graphs (Lin et al., 2015; Xie et al., 2018; Jia et al., 2019). Zhang et al. (2022b) show that contrastive learning between the information of neighboring triples of the head and tail entities is more effective because of the triple-level contrasting instead of entity or graph-level, leading to superior performance compared to all path-based methods. Clustering methods (Ge et al., 2020) are also used to partition noise from triples, and an active learning-based classification model is proposed to detect and repair dirty data. While these methods consider local information, our work also accounts for semantic information and the global

information of the knowledge graph to guide noise detection, better mitigating the impact of noise on local information. Regarding direct noise detection in CSKGs, Romero and Razniewski (2023) study the problem of mapping the open KB into the structured schema of an existing one, while our methods only use the CSKG to be denoised itself, not relying on any other CSKG.

## 2.2 Knowledge Graph Rule Mining

Another related line of work is knowledge graph rule mining, which is essential to our method. This task has received great attention in the knowledge graph completion. The first category of methods is Inductive Logical Programming (ILP) (Muggleton and Raedt, 1994), which uses inductive and logical reasoning to learn rules. On the other hand, AMIE (Galárraga et al., 2013) proposes a method of association rule mining, which explores frequently occurring patterns in the knowledge graph to extract rules and counts the number of instances supporting the discovered rules and their confidence scores. AMIE+ (Galárraga et al., 2015) and AMIE 3 (Lajus et al., 2020) further improve upon this method by introducing several pruning optimizations, allowing them to scale well to large knowledge graphs. SWARM (Barati et al., 2017) also introduces a statistical method for rule mining in large-scale knowledge graphs that focuses on both instance-level and schema-level patterns. However, it requires type information of entities, which is not available in the CSKG and, therefore, cannot be applied to CSKG. Recently, with the success of deep learning, the idea of ILP has been neuralized, resulting in a series of neural-symbolic methods. Neural LP (Yang et al., 2017) and DRUM (Sadeghian et al., 2019) both propose end-to-end differentiable models for learning first-order logical rules for knowledge graph reasoning. Despite the great success achieved by the combination of Recurrent Neural Network (RNN) (Schuster and Paliwal, 1997) with rule mining (Qu et al., 2021; Cheng et al., 2022, 2023), neuralized methods are intuitively hard to interpret due to the confidence scores output by neural networks. Furthermore, jointly learning rules and embedding has been proven to be effective (Guo et al., 2016), and iteratively learning between them can also promote the effectiveness of both (Guo et al., 2018; Zhang et al., 2019b). For noise detection in knowledge graphs, Belth et al. (2020) learn higher-order patterns based on subgraphs to help refine knowledge graphs, but it requires type information of node and hence cannot be applied to the CSKG.

## 2.3 Knowledge Graph Completion with Pretrained Language Models

Aside from specifically designed noise-detection methods, the line of works targeting KG completion can also be transferred to tackle noise-detection tasks. Previous research has shown that PLMs can achieve outstanding performance on KG completion tasks for both conventional KGs (Wang and Li, 2016; An et al., 2018; Yao et al., 2019; Wang et al., 2021b; Markowitz et al., 2022; Shen et al., 2022) and CSKGs (Su et al., 2022; Yasunaga et al., 2022) due to their ability to capture linguistic patterns and semantic information. However, two limitations still exist. First, performing edge classification using a PLM requires optimizing a large number of parameters on textual data that has been transformed from edges in CSKGs. Such fine-tuning is not only computationally expensive but also incapable of learning structural features in graphs, which are essential for accurately identifying and classifying edges. Second, recent studies (Safavi et al., 2021; Chen et al., 2023) have shown that language models, regardless of their scale, struggle to acquire implicit negative knowledge through costly language modeling. This makes them potentially vulnerable to noise detection tasks, as these noises typically belong to negative knowledge. Therefore, more sophisticated manipulations of the semantic information extracted by PLMs are needed to leverage them for noise detection tasks efficiently.

## 3 Problem Definition

**Noises in CSKG** Commonsense knowledge represents not only basic facts in traditional knowledge graphs but also the understanding possessed by most people (Liu and Singh, 2004), we evaluate whether a triple is a noise from two perspectives:

- *Truthfulness*: It should be consistent with objective facts. For example, (*London*, IsA, *city in France*) is not true because London is not in France but in England.
- *Reasonability*: It should align with logical reasoning and be consistent with cultural norms. For example, (*read newspaper*, MotivatedByGoal, *want to eat vegetables*) is not logically reasonable. The two nodes are

not directly related, and there is no clear relationship between them. Another example is that (*hippo*, AtLocation, *in kitchen*) violates our understanding and experience of reality because hippos are large mammals that are highly unlikely and unrealistic to be found in a kitchen.

If a triple fails to satisfy any of the aspects mentioned above, we define it as noise.

**CSKG Denoising** A CSKG can be represented as $G = (\mathcal{V}, \mathcal{R}, \mathcal{E})$, where $\mathcal{V}$ is a set of nodes, $\mathcal{R}$ is a set of relations, and $\mathcal{E} \subseteq \mathcal{V} \times \mathcal{R} \times \mathcal{V}$ is a set of triples or edges. Given a triple $(h, r, t) \in \mathcal{E}$ in a CSKG, we concatenate the language descriptions of $h$, $r$, and $t$ and determine whether this description conforms to commonsense. We note that each triple violates commonsense to a different degree, and we define noise detection as a ranking problem to standardize the evaluation process better. Thus, we model noise detection as a ranking process where a scoring function $f : \mathcal{E} \to \mathbb{R}$ indicates the likelihood of the triple being noisy.

## 4 The GOLD Method

Our proposed method GOLD comprises four components: triple encoder, global noise detector, local noise detector, and comprehensive evaluation scorer. An overview is presented in Figure 2. First, we leverage a PLM to encode the natural language descriptions of nodes and relations in CSKGs to obtain their sentence embeddings, thus further encoding the triples. When detecting noise, we evaluate the likelihood of a triple being noise from both a global and local perspective. From the global perspective, we aim to identify high-frequency patterns in the knowledge graph, as a small amount of noise is less likely to affect correct high-frequency patterns (Belth et al., 2020). To accomplish this, we employ rule mining to extract high-quality rules from the knowledge graph. From the local perspective, we adopt graph networks to aggregate the neighboring triple information around both the head and tail nodes of a given edge, allowing us to estimate if there is any correlation. Finally, based on these two aspects of detection, we obtain a comprehensive score indicating the noise level.

### 4.1 Triple Encoder

As we mentioned earlier, the nodes in CSKG are linguistic descriptions that are not restricted to any specific canonicalized form. If their semantic information is ignored, it will inevitably affect the accuracy of noise detection. Therefore, the Triple Encoder (**TE**) employs a PLM to encode the semantics of each node and relation. For instance, considering an example of triple $(h, r, t)$, their embeddings are defined as:

$$\boldsymbol{s}_h = \mathrm{LM}(h), \boldsymbol{s}_r = \mathrm{LM}(r), \boldsymbol{s}_t = \mathrm{LM}(t), \quad (1)$$

where LM is a frozen PLM that maps the input text to an embedding. To strike a balance between capturing the relationship between $h$, $r$, and $t$ and maintaining model efficiency, we opt an efficient RNN as our encoding method for the CSKG triples:

$$\boldsymbol{e}_h, \boldsymbol{e}_r, \boldsymbol{e}_t = \mathrm{RNN}(\boldsymbol{s}_h, \boldsymbol{s}_r, \boldsymbol{s}_t). \quad (2)$$

Then, we simply concatenate them together to get the representation of the triple $(h, r, t)$:

$$\mathbf{TE}(h, r, t) = [\boldsymbol{e}_h \| \boldsymbol{e}_r \| \boldsymbol{e}_t]. \quad (3)$$

### 4.2 Global Rule Mining

To detect noisy triples, scoring $(h, r, t)$ only from a local perspective, such as modeling the neighbors of $h$ and $t$, or analyzing the path from $h$ to $t$ may not be sufficient to eliminate the interference of noisy triples, as it is difficult to determine what is noise from local structures alone. In commonsense knowledge graphs, the noise ratio should not be excessively high. So, learning high-frequency patterns from a global perspective is likely to cover correct triples. In turn, patterns can guide us in identifying the noise data when detecting violations.

To incorporate the global information of the entire CSKG when determining the probability of a triple being noise, we use the method of rule mining to first extract high-frequency, high-confidence, and interpretable rules from the CSKG. Taking into account both the interpretability and efficiency of the model, we employ AMIE 3 (Lajus et al., 2020), a rule mining method based on the frequency of each pattern, to generate logical rules automatically with the following format:

$$r_h(x, y) \leftarrow r_{b_1}(x, z_1) \wedge \cdots \wedge r_{b_k}(z_{k-1}, y), \quad (4)$$

where $r_h(x, y)$ is rule head and $r_{b_1}(x, z_1) \wedge \cdots \wedge r_{b_k}(z_{k-1}, y)$ is rule body, $x$, $y$, $z_1$, ..., $z_{k-1}$ are nodes, $r_h$, $r_{b_1}$ ..., $r_{b_k}$ are relations. As depicted in

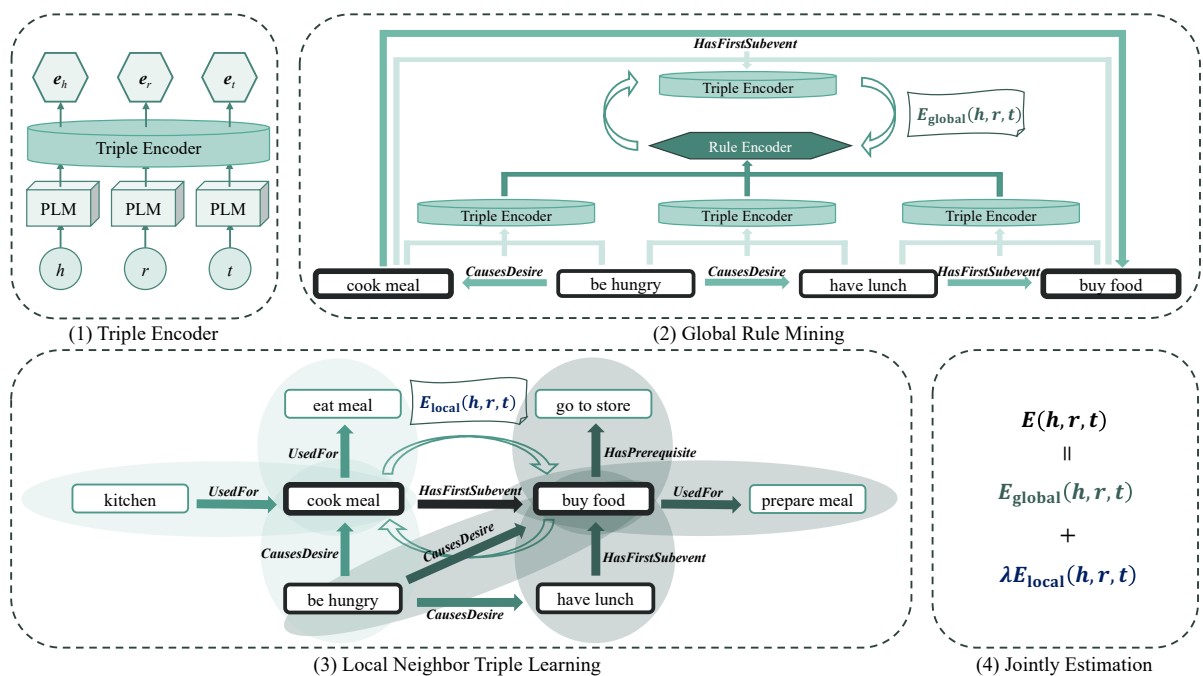

| | |
|---|---|
| (1) Triple Encoder | (2) Global Rule Mining |
| (3) Local Neighbor Triple Learning | (4) Jointly Estimation |

Figure 2: Overview of our GOLD framework. The example showing how to examine the noise level of the triple (*cook meal*, HasFirstSubevent, *buy food*) from both global and local perspectives are presented in the figure. The rule HasFirstSubevent$(x, y) \leftarrow$ CausesDesire$(z_1, x) \wedge$ CausesDesire$(z_1, z_2) \wedge$ HasFirstSubevent$(z_2, y)$ which is learned from the entire CSKG provides guidance to noise detection, while the neighboring triples of "*cook meal*" and "*buy food*" are used for aggregation as features for local structure learning.

Equation (4), the rule body consists of $k$ triples:

$$t_1 = (x, b_1, z_1), t_2 = (z_1, b_2, z_2), \cdots, t_k = (z_{k-1}, b_k, y). \quad (5)$$

To address the issue of poor generalization of mined rules due to sparsity in edges in CSKGs, we consider a rule body $\boldsymbol{r}_b$ as a sequence and employ an RNN as the neuralized Rule Encoder (**RE**) to generalize the rules:

$$\mathbf{RE}(\boldsymbol{r}_b) = \text{RNN}\left(\mathbf{TE}(t_1), \mathbf{TE}(t_2), \cdots, \mathbf{TE}(t_k)\right). \quad (6)$$

Specifically, for each relation as the rule head, we retain the top $k_{rules}$ rules with the highest confidence score given by AMIE 3 for training the rule encoder. In cases where there is no corresponding instance for a rule body, we fill all triples in the rule body with $(x, h, y)$ to align the energy scores of the other triples. And we believe that a well-generalized rule encoder can learn a representation that can explicitly infer the rule head $\boldsymbol{r}_h$, i.e., $(x, h, y)$. Hence, we align the dimensions of the outputs from **TE** and **RE** and define the energy function as follows:

$$E_{\text{global}}(h, r, t) = \sum_{(\boldsymbol{r}_b, \boldsymbol{r}_h)} \|\mathbf{RE}(\boldsymbol{r}_b) - \mathbf{TE}(\boldsymbol{r}_h)\|_2. \quad (7)$$

### 4.3 Local Neigboring Triple Learning

Structural information plays a significant role in enhancing performance for KG noise detection tasks. Most methods require that the relationship between two nodes should be equivalent to a translation between their embeddings (Xie et al., 2018; Zhang et al., 2022b). We relax this restriction and aim to determine some level of contextual correlation between two related nodes. As for the specific relation, our global rule mining component will learn its corresponding representation. To capture the contextual semantic information of the triples around nodes, we adopt Graph Attention Network (GAT) (Velickovic et al., 2018) to aggregate the information of the neighboring triples.

We use a transformation matrix $\boldsymbol{W} \in \mathbb{R}^{F \times d}$ to map the $i$-th triple $(h_i, r_i, t_i)$ to the embedding

$$\boldsymbol{v}_i = \boldsymbol{W}\left[\boldsymbol{e}_{h_i} || \boldsymbol{e}_{r_i} || \boldsymbol{e}_{t_i}\right] \quad (8)$$

where $F$ is the dimension of the latent space and $d$ is the embedding dimension of the triple, and perform the self-attention function $a : \mathbb{R}^F \times \mathbb{R}^F \to \mathbb{R}$ on the triples to get $w_{ij} = a\left(\boldsymbol{v}_i, \boldsymbol{v}_j\right)$, which indicates the context of the $j$-th triple to the $i$-th triple. To compute the attention of the neighboring

triples on the head and tail nodes, respectively, we define the neighboring triples of the node $e$ as $\mathcal{N}_e = \{(\tilde{h}, \tilde{r}, \tilde{t}) | \tilde{h} = e \vee \tilde{t} = e\}$, and then use the softmax function to normalize the coefficients:

$$
\begin{aligned}
\alpha_{ij(h)} &= \text{softmax}_{j(h)}(w_{ij(h)}) \\
&= \frac{\exp(w_{ij(h)})}{\sum_{k(h) \in \mathcal{N}_{h_i}} \exp(w_{ik(h)})}, \\
\beta_{ij(t)} &= \text{softmax}_{j(t)}(w_{ij(t)}) \\
&= \frac{\exp(w_{ij(t)})}{\sum_{k(t) \in \mathcal{N}_{t_i}} \exp(w_{ik(t)})},
\end{aligned}
\tag{9}
$$

where $\alpha_{ij(h)}$ represents the attention of the $j^{(h)}$-th triple on node $h_i$, while $\beta_{ij(t)}$ represents the attention of the $j^{(t)}$-th triple on node $t_i$. It is worth noting that the $j^{(h)}$-th triple is required to meet the condition of being a neighbor of node $h_i$, and similarly, the $j^{(t)}$-th triple must also be a neighbor of node $t_i$.

We use the normalized attention coefficients to calculate a linear combination of the corresponding embeddings, which then serves as the final output:

$$
\begin{aligned}
\boldsymbol{p}_i &= \sigma\left(\sum_{j(h) \in \mathcal{N}_{h_i}} \alpha_{ij(h)} \boldsymbol{v}_{j(h)}\right), \\
\boldsymbol{q}_i &= \sigma\left(\sum_{j(t) \in \mathcal{N}_{t_i}} \beta_{ij(t)} \boldsymbol{v}_{j(t)}\right),
\end{aligned}
\tag{10}
$$

where $\boldsymbol{p}_i$ is obtained from the perspective of the neighbors of node $h_i$, $\boldsymbol{q}_i$ is obtained from the perspective of the neighbors of node $t_i$, and $\sigma$ represents a nonlinearity.

We simply employ the Euclidean distance between them to measure the correlation between $h_i$ and $t_i$ and obtain the energy function of triple $(h_i, r_i, t_i)$ under local perception as follows:

$$
E_{\text{local}}(h_i, r_i, t_i) = \|\boldsymbol{p}_i - \boldsymbol{q}_i\|_2. \tag{11}
$$

### 4.4 Jointly Learning and Optimization

The overall energy function of each triple $(h, r, t)$ is obtained by combining the global and local energy functions. We have:

$$
E(h, r, t) = E_{\text{global}}(h, r, t) + \lambda E_{\text{local}}(h, r, t), \tag{12}
$$

where $\lambda$ is a hyperparameter.

We use negative sampling to minimize the margin-based ranking loss

$$
\mathcal{L} = \sum_{i^+ \in \mathcal{E}} \sum_{i^- \in \mathcal{E}_{i^+}} \max\left(0, \gamma + E(i^+) - E(i^-)\right), \tag{13}
$$

where $i^+$ represents a positive triple $(h, r, t)$, and $i^-$ represents a negative triple. We follow the setting of DistMult (Yang et al., 2015): a set of neg-

ative examples $\mathcal{E}_{i^+}$ is constructed based on $i^+$ by replacing either $h$ or $t$ with a random node $\tilde{e} \in \mathcal{V}$:

$$
\mathcal{E}_{i^+} = \{(\tilde{e}, r, t) | \tilde{e} \in \mathcal{V}\} \cup \{(h, r, \tilde{e}) | \tilde{e} \in \mathcal{V}\} - \mathcal{E}. \tag{14}
$$

## 5 Experimental Setup

### 5.1 Datasets

To evaluate the detection capability of denoising models, we follow the method introduced by Xie et al. (2018) to construct benchmark datasets for evaluation, which involves generating noise with manually defined sampling rules and injecting it back into the original CSKG. We select Concept-Net (Speer et al., 2017) and ATOMIC (Sap et al., 2019a) as two source CSKGs due to their manageable scale and diverse coverage of edge semantics, including various entities, events, and commonsense relations. Since these manually curated CSKGs do not contain noise naturally, we synthesize noise for each CSKG separately using meticulously designed rules, as done by Jia et al. (2019), that incorporate modifications on existing edges and random negative sampling. This approach, as demonstrated by Jia et al. (2019), ensures that the resulting noises not only maintain being highly informative, thus more challenging for the model to detect, but also stimulate several types of noise that may appear in real-world CSKGs. More details for noise synthesis are provided in Appendix A.1.

### 5.2 Evaluation Metrics

We use two common metrics to evaluate the performance of all methods.

**Recall@$k$.** Given that there are $k$ noisy triples in the dataset, we sort all triples by their score in descending order, where a higher score indicates a higher probability of being a noisy triple. We then select the top $k$ triples and calculate the recall rate:

$$
\text{Recall@}k = \frac{|\text{ Noisy Triples in the top-}k\text{ list }|}{k}. \tag{15}
$$

**AUC.** Area Under the ROC Curve (AUC) measures the probability that a model will assign a higher score to a randomly chosen noisy triple than a randomly chosen positive triple. A higher AUC score indicates a better performance.

### 5.3 Competing Methods

We compare our model with state-of-the-art models, which can be mainly divided into three categories: (i) structure embedding-based methods that are unaware of noise, including TransE (Bordes

| Model | ConceptNet | | | | | | ATOMIC | | | | | |
|---|---|---|---|---|---|---|---|---|---|---|---|---|
| | **N5** | | **N10** | | **N20** | | **N5** | | **N10** | | **N20** | |
| | R@5 | AUC | R@10 | AUC | R@20 | AUC | R@5 | AUC | R@10 | AUC | R@20 | AUC |
| TransE | .084 | .679 | .163 | .670 | .276 | .665 | .390 | .849 | .475 | .849 | .569 | .853 |
| DistMult | .118 | .656 | .187 | .652 | .283 | .653 | .425 | .841 | .490 | .835 | .551 | .840 |
| ComplEx | .160 | .733 | .248 | .720 | .364 | .718 | .460 | .842 | .531 | .841 | .581 | .839 |
| RotatE | .114 | .614 | .177 | .609 | .262 | .604 | .140 | .738 | .212 | .732 | .311 | .728 |
| CKRL | .150 | .693 | .231 | .701 | .342 | .694 | .317 | .787 | .411 | .795 | .497 | .794 |
| CAGED | .474 | .903 | .536 | .883 | .620 | .877 | .577 | .914 | .630 | .910 | .674 | .896 |
| KG-BERT | .601 | .925 | .680 | .936 | .750 | .939 | .714 | .936 | .782 | .953 | .813 | .951 |
| LASS (BERT-base) | .640 | .955 | .706 | .955 | .768 | .951 | .762 | .956 | .791 | .956 | .821 | .955 |
| LASS (BERT-large) | .689 | .959 | .750 | .963 | .804 | .961 | .757 | .957 | .792 | .957 | .827 | .957 |
| LASS (RoBERTa-base) | .665 | .961 | .709 | .958 | .775 | .955 | .775 | .961 | .802 | .960 | .831 | .959 |
| LASS (RoBERTa-large) | _.730_ | _.971_ | _.785_ | _.973_ | _.831_ | _.971_ | _.780_ | _.964_ | _.814_ | _.964_ | _.844_ | _.963_ |
| GOLD (RoBERTa-base) | .831 | .982 | .847 | .980 | .866 | .974 | .861 | .964 | .880 | .965 | .887 | .958 |
| GOLD (RoBERTa-large) | .828 | .985 | .841 | .978 | .868 | .977 | .864 | .968 | .880 | .962 | .900 | .968 |
| GOLD (DeBERTa-v3-base) | .839 | .979 | **.861** | .980 | .875 | .975 | .862 | .965 | .873 | **.967** | .884 | .959 |
| GOLD (DeBERTa-v3-large) | .823 | .973 | .850 | .975 | .863 | .968 | .849 | .962 | .863 | .958 | .880 | .961 |
| GOLD (Sentence-T5-base) | .838 | .983 | .852 | .981 | .870 | .975 | .863 | .959 | **.890** | .964 | .896 | .958 |
| GOLD (Sentence-T5-xl) | .822 | .982 | .836 | .979 | .858 | .973 | .862 | .960 | .880 | .962 | .891 | .960 |
| GOLD (Sentence-T5-xxl) | **.842** | **.985** | .859 | **.981** | **.878** | **.979** | **.872** | **.969** | .887 | .966 | **.901** | **.974** |

Table 1: Comparison of the effectiveness of different methods. We highlight that our proposed GOLD model outperforms all baselines across six data sets and both metrics. We denote the best results in **bold**, while the best result among the competing methods is marked with an underline. Further analysis is provided in Section 6.1.

et al., 2013), DistMult (Yang et al., 2015), ComplEx (Trouillon et al., 2016), and RotateE (Sun et al., 2019); (ii) embedding-based methods that are aware of noise, including CKRL (Xie et al., 2018) and CAGED (Zhang et al., 2022b); (iii) language model-based methods that encode both semantic and structural embeddings and are unaware of noise, including KG-BERT (Yao et al., 2019) and LASS (Shen et al., 2022). KGist (Belth et al., 2020) as a rule-based method requires node type information, which is unavailable in the CSKG, making it infeasible to use as a baseline. More detailed descriptions are in Appendix A.2.

## 5.4 Implementation Details

We leverage three families of PLMs from the Huggingface Library (Wolf et al., 2020) to build our GOLD framework, including RoBERTa (Liu et al., 2019), DeBERTa-v3 (He et al., 2023), and Sentence-T5 (Ni et al., 2022). Detailed variants of these PLMs are included in Table 1. We train GOLD with an Adam (Kingma and Ba, 2015) optimizer, with the learning rate set to 1e-3. The default number of training epochs is 10, with a margin $\gamma$ of 5 and a rule length set to 3. Additionally, we conduct a grid search for $\lambda$, ranging from 0 to 1, to find the best hyperparameter for $k_{rules}$ from 0 to 500. Further information regarding the implementation is discussed in Appendix A.3.

## 6 Experiments and Analyses

### 6.1 Main Results

The performance of all models on the six datasets in the noise detection task is shown in Table 1. In general, GOLD can detect noise in CSKG more accurately, outperforming all baseline methods by a large margin. Unlike baseline models based on language models, whose performance significantly increases with the size of the language model, our GOLD method consistently surpasses the baseline across different language model backbones with small performance variation. Specifically, when using the RoBERTa family of language models, our GOLD method achieves an average accuracy improvement of 8.64% and 8.50% compared to LASS methods on the ConceptNet and ATOMIC dataset series, respectively. Among the language models we use, the Sentence-T5-xxl model exhibits the best overall performance, with the highest accuracy improvement over 10.14% and 9.17% on the ConceptNet and ATOMIC dataset series, respectively, compared to the baseline. Additionally, the AUC score also improves by 1.02% and 0.62%.

### 6.2 Ablation Study

In this section, we conduct an ablation study on the ConceptNet-N10 dataset to evaluate the contribution of each component in our proposed model.

| Model | Recall@$k$ | AUC |
|---|---|---|
| GOLD (Sent-T5-xxl) | 0.859 | 0.981 |
| w/o LM | 0.810($\downarrow$ 5.7%) | 0.968($\downarrow$ 1.3%) |
| w/o $E_{\text{global}}$ | 0.826($\downarrow$ 3.8%) | 0.971($\downarrow$ 1.0%) |
| w/o $E_{\text{local}}$ | 0.599($\downarrow$ 30.1%) | 0.925($\downarrow$ 5.7%) |

Table 2: Ablation study results comparing the performance of GOLD with and without each component on the ConceptNet-N10 dataset.

The results of this study are presented in Table 2. Overall, we observe that removing any of the components results in varying degrees of performance degradation, emphasizing the essentiality of each component in our GOLD model.

**Influence of Language Model**   We remove the PLM from the triple encoder and use random embeddings to encode the information of nodes and relations, obtaining the embeddings $s_h, s_r, s_t$ in Equation (1). This results in a 5.7% decrease in the model's accuracy and a 1.3% decrease in AUC, indicating that the PLM indeed contributes to understanding the semantic information of nodes. It is worth noting that even after removing the language model, the accuracy and AUC still outperform all competing methods.

**Influence of Global Rule Mining**   We remove the global rule encoder, which results in a 3.8% decrease in accuracy and a 1.0% decrease in AUC, implying the important role of the rule encoder in guiding noise detection. Furthermore, as we train the rule encoder using the top $k_{rules}$ rules with the highest confidence score for each relation from the rules mined by AMIE 3, we test the impact of different values of $k_{rules}$ on the accuracy using three datasets from the ConceptNet series. We vary $k_{rules}$ among $\{100, 200, 300, 400, 500\}$. The results are shown in Figure 3. We observe that when the noise level is relatively low, i.e., in the N5 dataset, $k_{rules} = 200$ achieves the best performance, and adding more rules degrades the model's performance. However, increasing the number of rules improves the model's performance to some extent when the noise level is high, such as in the N10 and N20 datasets. We analyze that this is because as the noise level increases, learning local information becomes more prone to being misled. Hence, more rules are needed to provide global guidance.

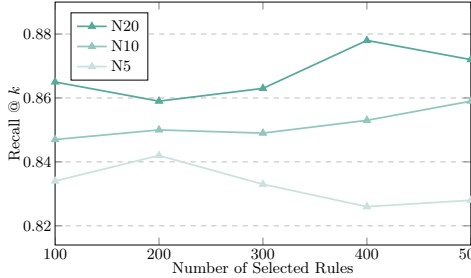

Figure 3: Accuracy of noise detection vs $k_{rules}$, the number of selected logical rules on ConceptNet series.

| Model | Recall@$k$ | AUC |
|---|---|---|
| GOLD (Sentence-T5-xxl) | 0.869 | 0.982 |
| GPT-3.5 (text-davinci-003) | 0.273 | 0.685 |
| ChatGPT (gpt-3.5-turbo) | 0.263 | 0.734 |

Table 3: Performance comparison on a randomly sampled ConceptNet-N10 dataset.

**Influence of Local Neighbor Learning**   Moreover, we remove the local neighbor information learning component, resulting in a significant decrease of 30.1% in accuracy and 5.7% in AUC, demonstrating the crucial role of neighboring triple information in noise detection. More comprehensive ablation studies are in Appendix C.

### 6.3   Comparison with ChatGPT

Recent breakthroughs in Large Language Models (LLMs), such as GPT-3.5 (Brown et al., 2020; Ouyang et al., 2022) and ChatGPT (OpenAI, 2022), have demonstrated remarkable performance across a diverse range of NLP tasks (Chan et al., 2023; Qin et al., 2023). In light of this, we benchmark these LLMs on our defined noise detection task to establish another competitive baseline for comparison. To accomplish this, we randomly select 1,000 triples from our poisoned ConceptNet-N10 CSKG and ask the LLMs to rank them by iteratively comparing two triples and merge-sorting them (more detailed information in Appendix B). This evaluation setting ensures that the LLMs follow an objective that is mostly identical to GOLD. The results, as shown in Table 3, indicate that both LLMs perform significantly poorly on our task, leaving a substantial gap compared to GOLD. One possible explanation is that these LLMs operate in a zero-shot setting and lack prior knowledge of noisy knowledge contained in CSKGs. This highlights the significance of GOLD, which exhibits a keen sensitivity to noise in CSKGs through fine-tuning.

## 6.4 Downstream Benefits of Denoising CSKG

We finally validate the effectiveness of our proposed noise detection framework by investigating whether eliminating noise from ATOMIC$^{10x}$ would yield extrinsic benefits for downstream tasks, specifically, zero-shot commonsense Question-Answering (QA) (Ma et al., 2021). This task involves performing QA on commonsense benchmarks, such as Abductive NLI (aNLI; Bhagavatula et al., 2020), CommonsenseQA (CSQA; Talmor et al., 2019), PhysicalIQA (PIQA; Bisk et al., 2020), SocialIQA (SIQA; Sap et al., 2019b), and WinoGrande (WG; Sakaguchi et al., 2021), without accessing their respective training data. Ma et al. (2021) proposed a technique that fine-tunes a PLM on synthetic QA pairs constructed from CSKGs, which has been further improved by Kim et al. (2022) using modularized transfer learning and Wang et al. (2023a) with conceptualizations (Wang et al., 2023b). Specifically, the head node and relation of an edge are transformed into a question using natural language templates, and the tail node serves as the ground-truth answer. Distractors are tails of other edges sampled from the same CSKG whose head node does not share common keywords with the question. A PLM is then fine-tuned on such synthetic QA entries using marginal ranking loss to serve as a general QA model. To this extent, we keep the QA synthesis protocol and model training process fixed and ablatively study the role of leveraging different CSKGs, in our case, raw ATOMIC$^{10x}$ and noise-cleaned ATOMIC$^{10x}$. We use accuracy as the evaluation metric and trained three QA models separately on (1) the original ATOMIC$^{10x}$, (2) ATOMIC$^{10x}$ denoised with LaSS, and (3) ATOMIC$^{10x}$ denoised with GOLD, where the former two served as the baselines. The results are reported in Table 4. We observe that cleaning ATOMIC$^{10x}$ with GOLD outperforms both baselines on average, indicating that denoising CSKG is potentially useful for automatically generated CSKGs and that GOLD is superior to other noise detection frameworks on real-world CSKGs.

## 6.5 Case Study

We present specific case studies on the mined logical rules and detected noises in the real large-scale CSKG in Appendix D. Those cases directly show the effectiveness of our proposed method.

| Denoising | aNLI | CSQA | PIQA | SIQA | WG | Avg. |
|---|---|---|---|---|---|---|
| N/A | **74.0** | 65.4 | 73.8 | **59.5** | **73.9** | 69.3 |
| LaSS | 71.8 | 65.8 | 77.7 | 57.4 | 67.3 | 68.0 |
| GOLD | 72.2 | **69.6** | **79.0** | 58.8 | 71.5 | **70.3** |

Table 4: Zero-shot evaluation results (%) on five benchmarks for QA models trained on the original/denoised ATOMIC$^{10x}$. N/A stands for not using any denoising technique, and Avg. refers to average.

## 7 Conclusions

In this paper, we propose GOLD, a noise detection framework leveraging the power of language models, global rules, and local structural information. This method is motivated by the fact that nodes in CSKGs are in free-text format, and correct patterns are unlikely to be drowned out by noise. Experimental results indicate that our method achieves state-of-the-art performances in CSKG noise detection tasks. This method shows promising directions for automatically obtaining a large-scale CSKG with minimal noise, as well as effectively representing knowledge for downstream tasks.

## Limitations

In our experiments, we follow the approach of previous noise detection literature (Xie et al., 2018; Jia et al., 2019) and inject synthesized noise back into the original CSKGs. Although this noise injection technique has been deemed reliable in previous works, further investigation is necessary to verify its rigor in the field of commonsense reasoning. This is because such noise can typically be classified as negative commonsense knowledge, which, as suggested by Chen et al. (2023), should be verified by whether it can be grounded as negative knowledge. Alternatively, we could inject noise from the perspective of graph attacks (Zhang et al., 2019a) to increase the difficulty of noise detection and improve the model's robustness.

## Ethics Statement

This paper introduces GOLD, a novel denoising framework for CSKG noise detection that is both global and local-aware. The experiments presented in this paper utilize open-source datasets, including ConceptNet, ATOMIC, ATOMIC$^{10x}$, and five commonsense question-answering benchmarks. The crowdsourced datasets, such as ConceptNet, ATOMIC, and the five commonsense question-answering benchmarks, have been manually cu-

rated and further processed to ensure that they are anonymized and desensitized. The experiments align with their intended usage, which is for research purposes. Additionally, while ATOMIC$^{10x}$ is generated using language models, its prompt engineering ensures that no harmful content is generated, which has been verified by manual inspection (West et al., 2022). Therefore, to the best of the authors' knowledge, we believe that GOLD introduces no additional risk.

## Acknowledgements

The authors would like to thank the anonymous reviewers for their valuable feedback. The authors of this paper were supported by the NSFC Fund (U20B2053) from the NSFC of China, the RIF (R6020-19 and R6021-20), and the GRF (16211520 and 16205322) from RGC of Hong Kong. We also thank the UGC Research Matching Grants (RMGS20EG01-D, RMGS20CR11, RMGS20CR12, RMGS20EG19, RMGS20EG21, RMGS23CR05, RMGS23EG08).

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

# Appendices

## A   Experimental Setup Details

### A.1   Datasets

**ConceptNet**   ConceptNet, or CN-100K, was first proposed by (Li et al., 2016). It contains Open Mind Common Sense (OMCS) in the ConceptNet 5 dataset. CN-82K dataset (Wang et al., 2021a) is a uniformly sampled version of the CN-100K dataset.

**ATOMIC**   ATOMIC contains over 300K everyday commonsense knowledge nodes, organized as *if-then* relations. It proposes nine types of *if-then* relations to distinguish various aspects of events, such as causality, intents, and mental states. Malaviya et al. constructed a dataset from ATOMIC for the task of CSKG completion.

In our experiments, we follow Wang et al. (2021a) to use CN-82K and ATOMIC. Unlike CSKG completion settings, we merge the train, valid, and test split to get training and testing sets because noise detection is a ranking task requiring training and testing on the entire knowledge graph. To introduce noisy triples, we follow Xie et al. (2018) and Jia et al. (2019) to add noisy triples to these two datasets separately manually. Specifically, the noise we generate is divided into four parts, with a probability of $1/4$ for randomly generating a new triple $(\hat{h}, \hat{r}, \hat{t})$ where $\hat{h}, \hat{t} \in \mathcal{V}, \hat{r} \in \mathcal{R}$, and probabilities of $1/4$ each for modifying the head node, relation, or tail node of an existing triple. When modifying an existing triple, we randomly sample a ground truth triple $(h, r, t) \in \mathcal{E}$ from the CSKG and then replace one of its components with a randomly chosen node $\hat{h}, \hat{t} \in \mathcal{V}$, or relation $\hat{r} \in \mathcal{R}$, to create a new triple $(\hat{h}, r, t), (h, \hat{r}, t)$ or $(h, r, \hat{t})$. The process of generating noisy triples requires ensuring that they do not exist in the original CSKG. Taking (*hotel room*, UsedFor, *temporary residence*) from ConceptNet and (*John works long hours*, xIntent, *to make more money*) from ATOMIC as examples, Table 6 presents several examples of noise generated by replacing the head node, relation, and tail nodes, as well as examples of newly generated triples. It can be observed that these noises are still informative and theoretically challenging to detect, aligning with our previous definition of noises in CSKG in Section 3. Hence, we believe that the noise generated through the above method is effective for model training. The statistical information for the datasets is presented in Table 5.

### A.2   Competing Methods

We compare GOLD with three categories of algorithms, beginning with four structure embedding-based methods that are unaware of noise. Here, $h, r, t$ represent the embeddings of the head entity, relation, and tail entity, respectively.

- **TransE** (Bordes et al., 2013) The score function is $\|h + r - t\|$, where $h, r, t \in \mathbb{R}^d$.

- **DistMult** (Yang et al., 2015) The score function is $\langle r, h, t \rangle$, where $\langle \cdot \rangle$ denotes the generalized dot product, and $h, r, t \in \mathbb{R}^d$.

- **ComplEx** (Trouillon et al., 2016) The score function is $\Re\left(\langle r, h, \bar{t} \rangle\right)$, where $h, r, t \in \mathbb{C}^d$.

- **RotatE** (Sun et al., 2019) The score function is $\|h \circ r - t\|^2$, where $\circ$ denotes the Hadamard product, and $h, r, t \in \mathbb{C}^d$.

Next, we consider two embedding-based methods that capture noise using local information:

- **CKRL** (Xie et al., 2018) They introduce the triple confidence and path confidence to conventional translation-based methods for knowledge representation learning.

- **CAGED** (Zhang et al., 2022b) They propose a contrastive learning framework to capture noise by aggregating triple information around the head and tail entities while also learning the traditional translation embedding.

We also evaluate our methods against fine-tuned language models:

- **KG-BERT** (Yao et al., 2019) They first propose concatenating the triples into textual descriptions and transforming the representation learning into a triplet classification problem. We evaluate the performance of noise detection by using scores designed for classification.

- **LASS** (Shen et al., 2022) They propose a joint language semantic and structure embedding for knowledge graph completion. We also use the scores designed for triplet classification to evaluate the performance. Experimental results from their paper demonstrate that their

| Dataset | Nodes | Relations | Original Triples | Noisy Triples | Avg. Degree | Avg. Words |
|---------|-------|-----------|------------------|---------------|-------------|------------|
| ConceptNet-N5 | | | | 5,120 | | |
| ConceptNet-N10 | 78,339 | 34 | 102,400 | 10,240 | 1.31 | 2.85 |
| ConceptNet-N20 | | | | 20,480 | | |
| ATOMIC-N5 | | | | 39,297 | | |
| ATOMIC-N10 | 304,439 | 9 | 762,230 | 78,595 | 2.50 | 4.47 |
| ATOMIC-N20 | | | | 157,190 | | |

Table 5: Statistical information for six datasets. Avg. Degree represents the average degree of each node and Avg. Words represents the average number of words in the text description of each node.

| Type | Head | Relation | Tail |
|------|------|----------|------|
| Replacing $h$ with $\hat{h}$ | *playground equipment* 
 *John has trouble falling asleep* | UsedFor 
 xIntent | *temporary residence* 
 *to make more money* |
| Replacing $r$ with $\hat{r}$ | *hotel room* 
 *John works long hours* | NotCapableOf 
 oEffect | *temporary residence* 
 *to make more money* |
| Replacing $t$ with $\hat{t}$ | *hotel room* 
 *John works long hours* | UsedFor 
 xIntent | *prepare food to eat* 
 *lose money or resources* |
| New Triple | *plastic fork* 
 *John drinks coffee* | CapableOf 
 oEffect | *buy food* 
 *to go to the movie theatre* |

Table 6: Examples of four manually generated types of noise. The ground truth triples modified in the examples are (*hotel room*, UsedFor, *temporary residence*) from ConceptNet and (*John works long hours*, xIntent, *to make more money*) from ATOMIC. The modified parts are indicated by underlines.

model outperforms other PLM-based methods in triplet classification tasks. Hence, we select it as our baseline. In particular, their model is tested on four backbones, namely BERT-base, BERT-large, RoBERTa-base, and RoBERTa-large. We also conduct experiments on these four backbones.

### A.3 Implementation Details

For the embedding-based baseline models, we use the implementation from OpenKE(Han et al., 2018). For the rest, we use the released code corresponding to each paper to perform experiments. In order to align the performance of different models, we set the dimension of all embeddings apart from language models to 100, the number of negative samples to 1, and the batch size to 256. Our model also follows these settings. For the remaining hyperparameters of baseline models, we follow the settings proposed in the original paper and perform a grid search when modifications are necessary.

### B Details of the Zero-shot Noise Detection

ChatGPT cannot directly sort a large number of triples, so we implement a merge sort in Algorithm 1 to sort the triples in descending order of their noise level. When comparing the order of two triples, we draw inspirations from Wang et al.

(2023c) and call the ASKCHATGPT function to employ ChatGPT to choose which triple is more likely to be noisy from two triples. Inspired by chain-of-thought (CoT) prompting (Wei et al., 2022), we guide ChatGPT in the prompt to first provide the specific reasoning process and then compel it to provide the answer. The prompt used for comparing which of the two triples is more likely to be noise is listed in Table 7. We use OpenAI's API[1] to prompt ChatGPT and retrieve its response.

| Prompt |
|--------|
| Given two triples from a knowledge graph: $(h_1, r_1, t_1), (h_2, r_2, t_2)$. Which one is more likely to be wrong? Show me the reason first, and then print the wrong triple. You are forced to make a decision. |

Table 7: A natural language prompt used to guide ChatGPT to compare two triples and determine which one is more likely to be noise. Entries in italics will be replaced by actual triples. The last sentence mandates ChatGPT to choose which of the two triples is more likely to be noise.

### C Full Results of Ablation Study

In this section, we provide a comprehensive supplementary ablation study. The results of all exper-

---

[1]https://chat.openai.com/

| Model | ConceptNet | | | | | |
| | N5 | | N10 | | N20 | |
| | Acc | AUC | Acc | AUC | Acc | AUC |
| GOLD (Sent-T5-xxl) | **.842** | **.985** | **.859** | **.981** | **.878** | **.979** |
| w/o PLM | .779(↓ 7.5%) | .970(↓ 1.5%) | .810(↓ 5.7%) | .968(↓ 1.3%) | .834(↓ 5.0%) | .963(↓ 1.6%) |
| w/o $E_{\text{global}}$ | .791(↓ 6.1%) | .974(↓ 1.1%) | .826(↓ 3.8%) | .971(↓ 1.0%) | .862(↓ 1.8%) | .974(↓ 0.5%) |
| w/o $E_{\text{local}}$ | .478(↓ 43.2%) | .915(↓ 7.1%) | .599(↓ 30.3%) | .925(↓ 5.7%) | .652(↓ 25.7%) | .907(↓ 7.4%) |
| w/ $E_{\text{translation}}$ | .841(↓ 0.1%) | .986(↑ 0.1%) | .856(↓ 0.4%) | .983(↑ 0.2%) | .867(↓ 1.3%) | .971(↓ 0.8%) |

| Model | ATOMIC | | | | | |
| | N5 | | N10 | | N20 | |
| | Acc | AUC | Acc | AUC | Acc | AUC |
| GOLD (Sent-T5-xxl) | **.872** | **.969** | **.887** | **.966** | **.901** | **.974** |
| w/o PLM | .779(↓ 10.7%) | .927(↓ 4.3%) | .801(↓ 9.7%) | .928(↓ 3.9%) | .822(↓ 8.8%) | .929(↓ 4.6%) |
| w/o $E_{\text{global}}$ | .859(↓ 1.5%) | .960(↓ 0.9%) | .874(↓ 1.5%) | .961(↓ 0.5%) | .884(↓ 1.9%) | .955(↓ 2.0%) |
| w/o $E_{\text{local}}$ | .656(↓ 24.8%) | .931(↓ 3.9%) | .699(↓ 21.2%) | .930(↓ 3.7%) | .747(↓ 17.1%) | .926(↓ 4.9%) |
| w/ $E_{\text{translation}}$ | .868(↓ 0.5%) | .962(↓ 0.7%) | .886(↓ 0.1%) | .965(↓ 0.1%) | .901(↓ 0.0%) | .970(↓ 0.4%) |

Table 8: The comprehensive ablation study results comparing the impact of each component on the results on all six datasets. Additionally, we verify the effect of adding the translation-based energy function on the results.

---

**Algorithm 1** Merge Sort guided by ChatGPT

**Input**: A triple list $L$
**Output**: A tiple list sorted from high to low according to the noise level
**Function**: MERGESORT($L$)
1: $h \leftarrow |L|/2$
2: $L_{left} \leftarrow$ MERGESORT($L[1, 2, \cdots, h]$)
3: $L_{right} \leftarrow$ MERGESORT($L[h + 1, \cdots |L|]$)
4: $i \leftarrow 1$
5: $j \leftarrow 1$
6: **for** $k \leftarrow 1$ **to** $|L|$ **do**
7:    **if** $i > h$ **then**
8:       $L[k] \leftarrow L_{right}[j]$
9:       $j \leftarrow j + 1$
10:   **else if** $j > h$ **then**
11:      $L[k] \leftarrow L_{left}[i]$
12:      $i \leftarrow i + 1$
13:   **else if** ASKCHATGPT($L_i, L_j$) = $L_i$ **then**
14:      $L[k] \leftarrow L_{left}[i]$
15:      $i \leftarrow i + 1$
16:   **else**
17:      $L[k] \leftarrow L_{right}[j]$
18:      $j \leftarrow j + 1$
19:   **end if**
20: **end for**
21: **return** $L$

---

iments conducted on the six datasets are listed in Table 8.

**Influence of Language Model** By removing the PLM from the triple encoder, we observe an average decrease of 6.1% in accuracy on the Concept-Net series datasets and an average decrease of 9.7% on the ATOMIC series datasets. This indicates that PLM has a greater impact on the accuracy of the ATMOIC datasets, as the average number of words per node in ATOMIC is much higher than that in

ConceptNet. Therefore, PLM plays a more crucial role in capturing semantic information.

**Influence of Global Rule Mining** After eliminating the global rule encoder, the accuracy of the ConceptNet series and ATOMIC series datasets decreases by 3.9% and 1.6%, respectively. Our analysis suggests that the lower number of relations in the ATOMIC datasets, only 9 compared to 34 in the ConceptNet datasets, results in a significantly lower number of learnable rules compared to the ConceptNet. As a result, the global rule encoder provides limited assistance in the ATOMIC datasets, and its contribution is not as significant as in the ConceptNet datasets.

**Influence of Local Neighbor Learning** The local neighbor learning component exhibits the highest contribution across all datasets, as evidenced by the average accuracy drops of 33.1% and 21.0% in accuracy, as well as 6.7% and 4.2% in AUC after its removal on ConceptNet series and ATOMIC series datasets, respectively. We believe that the reason why this component has a smaller impact on the ATOMIC datasets is still due to the limited number of relations, leading to a less diverse set of information learned from the neighboring triple information.

**Influence of Translation Assumption** We attempt to investigate whether the model would benefit from the incorporation of a translation assumption, such as the $h + r \approx t$ relation in TransE (Bor-

| Rules |
|---|
| $\mathtt{IsA}(x, y) \leftarrow \mathtt{IsA}(x, z_1) \wedge \mathtt{DefinedAs}(z_1, z_2) \wedge \mathtt{IsA}(z_2, y)$ |
| $\mathtt{CapableOf}(x, y) \leftarrow \mathtt{HasA}(x, z_1) \wedge \mathtt{PartOf}(z_1, z_2) \wedge \mathtt{CapableOf}(z_2, y)$ |
| $\mathtt{NotDesires}(x, y) \leftarrow \mathtt{DefinedAs}(x, z_1) \wedge \mathtt{CreatedBy}(z_1, z_2) \wedge \mathtt{NotDesires}(z_2, y)$ |
| $\mathtt{HasPrerequisite}(x, y) \leftarrow \mathtt{HasPrerequisite}(x, z_1) \wedge \mathtt{MadeOf}(z_1, z_2) \wedge \mathtt{HasPrerequisite}(z_2, y)$ |

Table 9: Examples of the most frequent rules mined from the ConceptNet-N10 dataset.

| Head | Relation | Tail |
|---|---|---|
| *X is a good friend with Y* | isAfter | *X is a little weird* |
| *X refuses to take a pen* | oEffect | *are glad to see X* |
| *X steals Y's breakfast* | oNeed | *to leave X's room* |
| *X catches a stranger* | isAfter | *X checks the weather forecast* |
| *X flies to Washington* | oEffect | *become a politician* |
| *X falls down again* | isAfter | *X's mother gives X a hug* |
| *X seems to stay well* | oWant | *to see X get sick* |
| *X assumes that Y is a nice person* | isAfter | *Y has done something that upsets X* |
| *X studies hard for his exam* | oNeed | *to cheat on the exam* |
| *X is afraid of getting in trouble* | oNeed | *to tell X that he or she could get in trouble* |

Table 10: Example of noise detected in the $\textsc{Atomic}^{10\text{x}}$ CSKG.

des et al., 2013), where $\boldsymbol{h}, \boldsymbol{r}, \boldsymbol{t}$ represents the embedding of the head entity, relation, and tail entity respectively. Inspired by this, we also integrate an energy function based on the translation assumption into our approach. We design the energy function for the translation part as follows:

$$E_{\text{translation}}(h, r, t) = \|\boldsymbol{e}_h + \boldsymbol{e}_r - \boldsymbol{e}_t\|_2. \quad (16)$$

By adding Equation (16) to Equation (12), we obtain a new overall energy function as follows:

$$\begin{aligned} E(h, r, t) = {} & E_{\text{global}}(h, r, t) \\ & + \lambda E_{\text{local}}(h, r, t) \\ & + \lambda^{(t)} E_{\text{translation}}(h, r, t), \end{aligned} \quad (17)$$

where $\lambda$ and $\lambda^{(t)}$ are both hyperparameters. We perform a grid search for them between 0.001 to 1 and report the best results in Table 8. The experimental results indicate that the energy function based on the translation assumption in the form of Equation (16) cannot provide significant assistance to our model. The overall impact on precision is negative, with an average decrease of 0.4%. This suggests that our GOLD method does not need to rely on such translation assumption constraints when performing noise detection task. It can implicitly learn the relationship between nodes using the energy functions of the global and local parts.

## D   Case Studies

**Mined Logical Rules**   We list the most frequent rules mined from the ConceptNet-N10 dataset using AMIE 3 and present them in Table 9. We can observe that these rules are highly interpretable and not affected by mixed-in noise. Therefore, they can be treated as ground truth to validate the entire knowledge graph (Bai et al., 2023).

**Detected Noise**   We conduct our proposed GOLD method on the $\textsc{Atomic}^{10\text{x}}$ dataset and examine the triples with noise levels in the top 1%. We list ten specific examples that violate reasonability (see Section 3) in Table 10. The results show that our method can effectively extract noise triples from a large-scale CSKG.