# OpenReview forum: "Gold: A Global and Local-aware Denoising Framework for Commonsense Knowledge Graph Noise Detection"
_EMNLP/2023/Conference — EMNLP 2023 Findings_

### Official Review · Reviewer_UStH · 2023-08-04

**Soundness:** 2

**Excitement:**

3: Ambivalent: It has merits (e.g., it reports state-of-the-art results, the idea is nice), but there are key weaknesses (e.g., it describes incremental work), and it can significantly benefit from another round of revision. However, I won't object to accepting it if my co-reviewers champion it.

**Paper Topic And Main Contributions:**

The paper proposes a method named Global and Local-aware Denoising (GOLD) for the commonsense knowledge graph noise detection task. The proposed method uses the global detector to identify correct patterns, and utilizes a local detector to measure semantic similarity of triples. The proposed method achieves superior performace on several commonsen knowledge graphs, compared with previous baselines. The authors also conduct the ablation study to analyze the effect of each model component.

**Questions For The Authors:**

1. For the base model, such as RoBERTa or DeBERTa-v3, why do larger models achieve worse results?

**Reasons To Accept:**

1. The paper devises a method to detect the noise in commonsense KGs, which achieves better performace than previous baselines.

2. The paper is well-written and easy to follow.

**Reasons To Reject:**

1. The proposed approach is the combination of representation-based and rule mining-based methods, which lacks lacks novelty to some degree.

2. It would be better to investigate the noise ratio in the commonsense knwoledge graphs via sampling, since the commnonse KGs are usually constructed by experts or humans.

**Reproducibility:**

3: Could reproduce the results with some difficulty. The settings of parameters are underspecified or subjectively determined; the training/evaluation data are not widely available.

**Reviewer Confidence:**

4: Quite sure. I tried to check the important points carefully. It's unlikely, though conceivable, that I missed something that should affect my ratings.

---

> ### Author Rebuttal · Authors · 2023-08-29
>
> We express our gratitude to the Reviewer for providing us with valuable feedback. We would like to clarify the Reviewer’s concerns about novelty and address the comments related to the experiments.
>
> # Novelty of Our Method
>
> Thanks for raising the issue of novelty. We would like to clarify that our work goes beyond a mere combination of two existing approaches, and its novelty primarily stems from two key aspects: **(1) We introduce a new task: CSKG denoising, which can be applied to various CSKG construction and LLM distillation works. (2) We propose a novel approach that leverages both local and global information from textual data and graphs.**
>
> We would like to highlight that **we are the first to propose the denoising task for Commonsense Knowledge Graphs (CSKGs) and develop an effective framework for it**. In Section 6, we demonstrate that our proposed task **significantly improves downstream tasks** such as Commonsense Question Answering (CSQA). Specifically, our denoising models to our proposed task  (i.e., our proposed method to ATOMIC10X [1]) brings a 7.0% improvement in the PIQA dataset and a 6.4% improvement in the CSQA dataset when evaluating a zero-shot QA model trained on our denoised ATOMIC-10X. Those substantial improvements demonstrate that **this task holds great importance in the present context, especially with the emergence of more distillation works from large language models**. Unlike the traditional approach that relies on human annotation and critic filtering, our task provides a distinct option for cleaning distillation data. **Our denoising framework can be applied to any CSKG without the need for additional annotation or supervision signals, making it a powerful yet easy-to-use plugin for CSKGs**. Thus, our proposed task and accompanying method make a novel and impactful contribution to the NLP community.
>
> We would like to emphasize that our approach stands out by being the first to extract path-based logical rules and employ an RNN-based rule encoder to generalize them for denoising purposes. Figure 2 illustrates the joint training process, where a rule encoder in the global component compares triple and rule similarity. This information propagates to the bottom-level triple encoder, enabling simultaneous parameter updates in the local component. **This unique characteristic distinguishes our framework from a mere combination of existing modules**. Furthermore, we enhance our method by incorporating semantic information derived from Large Language Models (LLMs), which improves its adaptability to commonsense knowledge scenarios. Consequently, we firmly believe that our proposed approach goes beyond simply combining two strategies, but rather organically adapts them to the task of noise detection and the domain of commonsense reasoning.
>
> # Sample-based Analysis
> We would like to clarify that for the manually labeled CSKG, like ConceptNet and Atomic datasets, we would assume all of the data entries are correct. However, for the LLM-distilled large-scale CSKG, such as Atomic10X [1], a CSKB distilled from GPT3, there exists a certain amount of noise. As stated in the original paper [1], the manual evaluation of the sampled 3,000 triplets from the entire ATOMIC10X dataset resulted in an error rate of 18.7%.
> In line with your suggestion, we sample 100 triples from the top 1% with the highest anomaly probabilities in ATOMIC10X. We recruited three Ph.D. specialists in NLP, who can be considered experts in this topic, to annotate the data. The results revealed an average error rate of 55.7%. The IAA score is 77.3% calculated using pairwise agreement proportion, and Fleiss’s κ [2] is 0.54. The noise ratio extracted from the top 1% is significantly higher than the average error rate of the entire dataset, which indeed demonstrates the effectiveness of our approach. Due to time constraints, we will increase the sampling of the triples for manual evaluation in the subsequent revision.
>
>
> # Question about the Performance Deterioration of Large Models
> We would like to emphasize that we thoroughly reviewed the implementation details and did not find any issues with the implementation. Additionally, we would like to clarify that there is no guarantee that a large-size model will outperform a base-size model, particularly when the parameters are frozen. As an example in other zero-shot settings, in Table 1 of [3], it can be observed that deberta-large performs worse than deberta-base. Similarly, in Table 2 of [4], it is evident that the GPT2 model with 762M parameters is surpassed by the GPT2 model with 345M parameters.
>
>
>
> # Reference
> [1] Peter West, Chandra Bhagavatula, Jack Hessel, Jena D. Hwang, Liwei Jiang, Ronan Le Bras, Ximing Lu, Sean Welleck, and Yejin Choi. 2022. Symbolic knowledge distillation: from general language models to commonsense models. In Proceedings of the 2022 Conference of the North American Chapter of the Association for Computational Linguistics: Human Language Technologies, NAACL 2022, Seattle, WA, United States, July 10-15, 2022, pages 4602– 4625. Association for Computational Linguistics.
>
> [2] Fleiss, Joseph L. "Measuring nominal scale agreement among many raters." Psychological bulletin 76.5 (1971): 378.
>
> [3] Zhaowei Wang, Quyet V. Do, Hongming Zhang, Jiayao Zhang, Weiqi Wang, Tianqing Fang, Yangqiu Song, Ginny Wong, and Simon See. 2023. COLA: Contextualized Commonsense Causal Reasoning from the Causal Inference Perspective. In Proceedings of the 61st Annual Meeting of the Association for Computational Linguistics (Volume 1: Long Papers), pages 5253–5271, Toronto, Canada. Association for Computational Linguistics.
>
> [4] Bosselut, Antoine, Ronan Le Bras, and Yejin Choi. "Dynamic neuro-symbolic knowledge graph construction for zero-shot commonsense question answering." Proceedings of the AAAI conference on Artificial Intelligence. Vol. 35. No. 6. 2021.
>
>
> ***We genuinely appreciate your input and hope that our previous responses have adequately addressed your concerns and will help you improve your scores. Please let us know if you have other questions. We are more than happy to provide additional clarification and assistance.***

---

### Official Review · Reviewer_ys1w · 2023-08-07

**Soundness:** 3

**Excitement:**

3: Ambivalent: It has merits (e.g., it reports state-of-the-art results, the idea is nice), but there are key weaknesses (e.g., it describes incremental work), and it can significantly benefit from another round of revision. However, I won't object to accepting it if my co-reviewers champion it.

**Paper Topic And Main Contributions:**

The authors propose a novel denoising framework that acts over Commonsense Knowledge Graphs (CSKGs), usually characterized by spurious triples that reduce the quality of data collected in such a structure.

The introduced framework operates through semantic information, global rules and local structural information to distinguish noisy triples from real ones. Since CSKGs pair with Open KG (triples of text that define a fact), the authors describe an approach to identify the noise in the CSKG using entities' and triples' textual descriptions.

The evaluation stresses the link prediction task (LP) over two datasets through the Recall and Area Under the ROC Curve (AUC), focusing on how well such a model can detect noise. Hence, the authors test some downstream tasks to highlight the benefit of their approach.

**Reasons To Accept:**

The work targets a hot topic in the research, given the recent advancements, that is, identifying those textual facts producing noise in data collection.

The authors model an interesting idea to detect noise in such data by composing some existing components well adapted to the specified task. In doing so, they profoundly describe the state-of-the-art on which their proposal is positioned.

Also, the work presents a precise formalization and description of the overall architecture, although some essential definitions are left in the Appendixes.

**Reasons To Reject:**

The experiment run in this work can not properly state and evaluate the topic of interest treated by the authors. In fact, only the recall metrics and AUC could not completely evaluate such a system specifically designed to find noise. Specifically, precision would complete the overall model behaviour regarding noisy triples.

The evaluation task lacks relevant details to understand which configuration allows to stress the noise in the targeted dataset. Indeed, comparing classic LP models against the one proposed by the authors deceives the experiment, and the derived conclusion, validity.

Also, the Apenndixes are reserved to include related but optional work information. Instead, The authors have some central definitions, like the one regarding noisy triples and how to verify them, which is prominent in such a proposal. The Limitation section does not highlight the main limitations of this work but emphasises some future challenges (not the correct spot).

The research questions are not explicitly specified, as well as the main contributions, leaving the reader dazed about what the authors precisely want to achieve and how.

The downstream tasks bring no added value to the discussion. The authors could focus on the targeted task by proposing an in-depth and well-designed experimental configuration regarding noisy triples.

**Reproducibility:**

3: Could reproduce the results with some difficulty. The settings of parameters are underspecified or subjectively determined; the training/evaluation data are not widely available.

**Reviewer Confidence:**

4: Quite sure. I tried to check the important points carefully. It's unlikely, though conceivable, that I missed something that should affect my ratings.

---

> ### Author Rebuttal · Authors · 2023-08-29
>
> Thank you for your valuable feedback. In the following paragraphs, we would like to clarify your concerns one by one.
>
> # Evaluation Metric in Experiments
>
> Thanks for raising the issue of evaluation metric. We would like to clarify that our usage of AUC and Recall scores is consistent with previous works’ practice [2, 3]. As demonstrated in their works, such an evaluation is resilient and reliable in terms of detection rate and quality. Actually, in our paper, the reported Recall@k is equivalent to Precision@k, as defined below.
>
> $$Recall@k = \frac{| \text{Noises in Top } k \\% |}{| \text{Number of noises}|}$$
>
> $$Precision@k =  \frac{|  \text{Noises in Top } k \\% |}{| \text{Number of triples} \times k \\% |}$$
>
> Nevertheless, we also incorporate your suggestion and included the reporting of precision and recall when the noise ratio $k\\%$ increases accordingly.
>
> | Method               | P@1   | R@1   | P@2   | R@2   | P@3   | R@3   | P@4   | R@4   | P@5   | R@5   |
> |----------------------|-------|-------|-------|-------|-------|-------|-------|-------|-------|-------|
> | GOLD (T5-xxl)        | 0.999 | 0.200 | 0.996 | 0.398 | 0.982 | 0.589 | 0.927 | 0.741 | 0.842 | 0.842 |
> | LASS (roberta-large) | 0.903 | 0.181 | 0.880 | 0.352 | 0.857 | 0.514 | 0.800 | 0.640 | 0.730 | 0.730 |
> | KG-BERT              | 0.932 | 0.187 | 0.889 | 0.356 | 0.795 | 0.487 | 0.686 | 0.551 | 0.601 | 0.601 |
>
> The results above are obtained on the ConceptNet-N5 dataset, where the total noise injection ratio is 5%. It can be observed that our model significantly outperforms the baseline in terms of both recall and precision across these percentages.
>
> Due to the insufficient time, we only reproduce two strong baselines and our own method. We will follow your advice and incorporate a more comprehensive evaluation table in our final version.
>
> # Evaluation Details and Configuration
>
> Thanks for your question. We want to emphasize that our approach builds upon previous work (line 410), where noise is introduced into the original CSKG at different ratios. This method of noise injection has been widely adopted and justified by several subsequent studies. For implementation specifics, please refer to line 308 of the supplementary material file our_readers.py. By utilizing this robust evaluation setup, we can thoroughly assess the performance of our model across various noise ratios. Furthermore, in Section 6.2 and Appendix D, we present comprehensive results from ablation studies and elaborate on the contributions of each module to noise detection. Thus, we believe that our work, along with those previous works, possesses justifiable evaluation details and setups.
>
> # Inclusion of LP Models as Baselines
>
> We would like to clarify that our comparison with classic LP models is consistent with previous studies [1, 2, 3]. Additionally, we evaluated our method against two robust baselines [1, 3] that are specifically designed for denoising. The results demonstrate that our approach outperforms both baselines significantly. It is important to note that KGIst [2] was not included as a baseline in our work due to its requirement for node types, which is not a characteristic of the nodes in CSKG. Previous noise detection works also include the LP models as baselines for comparison, and we believe our baselines are comprehensive and competitive.
>
> # Definitions in Appendix
>
> Thank you for bringing up this issue. We would like to clarify our definition of noise is for the readers' better understanding. In our work, we define noise as knowledge that contradicts normal commonsense recognition in the human world (line 247). To enhance readability, we have also included two representative examples in Figure 1, which allows readers to grasp the meaning of noise in CSKG more easily. While being concise and comprehensive, we have also provided an additional explanation of noise within specific constraints, namely Truthfulness and Reasonability. It is important to note that our main goal is to formally propose the task of noise detection and emphasize that denoising improves performance in downstream tasks. The specific categorization of noise is an additional and optional aspect. We appreciate your suggestion and will consider moving it to the body text if space permits in our camera-ready version.
>
> # Limitation and Future Works
>
> Thank you for pointing out the limitation. We would like to clarify that our primary constraint lies in synthetic noise rather than human-verified noise, as mentioned in Lines 640-644. Additionally, we propose three potential approaches to address this limitation, which can serve as inspiration for future research. It is important to note that this limitation sheds light on the current constraints in our research and lays the groundwork for further investigations. Our suggested future works are not only practical but also hold value for other NLP researchers in the commonsense community. We believe that including this section will be advantageous for both our paper and the commonsense community. We appreciate your advice and will heed it by reducing the content pertaining to future challenges.
>
> # Research Questions and Main Contributions
>
> We would like to clarify that in Section 3, our task definition provides a comprehensive explanation of the research question. Our objective is to perform denoising on the Commonsense Knowledge Graph (CSKG) by assigning a probability score to each triple, indicating the likelihood of it being noise.
>
> Furthermore, our study presents the following key contributions and significant findings:
> * We propose the task of commonsense knowledge graph noise detection. (Section 3)
> * We propose Gold, a novel denoising framework for CSKGs, which outperforms all existing methods and LLMs. (Section 6.1, 6.3)
> * Experiment results demonstrate that Gold successfully detects noises in real-world CSKGs and benefits downstream zero-shot CSQA tasks. (Section 6.4, 6.5)
>
> # The Value of Downstream Task
>
> We want to emphasize that simply applying denoising techniques to a knowledge graph with synthetic noises is insufficient. To truly showcase the effectiveness of our denoising approach, it is crucial to validate its performance on downstream tasks. The value of the denoising method lies in its ability to enhance performance in these tasks. Our experiments on ATOMIC10X and zero-shot commonsense QA provide validation for our hypothesis, confirming the significance of our denoising approach.
>
> As Reviewer 68ji pointed out,
> ```
> The comprehensive experimental results demonstrate the effectiveness of the proposed method, especially compared to the LLM and deployed to downstream tasks.
> ```
> In addition, our paper includes a five-page appendix that contains comprehensive experimental details and in-depth analysis.
>
> # Reference
>
> [1] Ruobing Xie, Zhiyuan Liu, Fen Lin, and Leyu Lin. 2018. Does william shakespeare REALLY write hamlet? knowledge representation learning with confidence. In Proceedings of the Thirty-Second AAAI Conference on Artificial Intelligence, (AAAI-18), the 30th innovative Applications of Artificial Intelligence (IAAI 18), and the 8th AAAI Symposium on Educational Advances in Artificial Intelligence (EAAI-18), New Orleans, Louisiana, USA, February 2-7, 2018, pages 4954–4961. AAAI Press.
>
> [2] Caleb Belth, Xinyi Zheng, Jilles Vreeken, and Danai Koutra. 2020. What is normal, what is strange, and what is missing in a knowledge graph: Unified characterization via inductive summarization. In WWW’20: The Web Conference 2020, Taipei, Taiwan, April 20-24, 2020, pages 1115–1126. ACM / IW3C2.
>
> [3] Qinggang Zhang, Junnan Dong, Keyu Duan, Xiao Huang, Yezi Liu, and Linchuan Xu. 2022b. Contrastive knowledge graph error detection. In Proceedings of the 31st ACM International Conference on Information & Knowledge Management, Atlanta, GA, USA, October 17-21, 2022, pages 2590–2599. ACM.
>
> ***We hope the above responses address your concerns and will help you improve your scores. Please let us know if you have other questions. We are delighted to provide additional clarification and assistance.***

---

### Official Review · Reviewer_68ji · 2023-08-12

**Soundness:** 4

**Excitement:**

3: Ambivalent: It has merits (e.g., it reports state-of-the-art results, the idea is nice), but there are key weaknesses (e.g., it describes incremental work), and it can significantly benefit from another round of revision. However, I won't object to accepting it if my co-reviewers champion it.

**Paper Topic And Main Contributions:**

This paper aims to detect the noisy triples in the commonsense knowledge graph.

The main contribution is that they consider both the semantic information and the global and local information jointly when detecting noise. And this method obtains a significant performance improvement.

**Questions For The Authors:**

I am only concerned about the novelty of the implementation of global and local information modeling. Have they been separately proposed in existing studies, and you are simply combining them, or is there a new approach to combining their strategies? Your explanation would help me better understand your innovative contributions and potentially raise my rating to a 4.

**Reasons To Accept:**

1. A good motivation to utilize local and global information jointly and successfully implement it via a simple but effective modeling method.

2. The comprehensive experimental results demonstrate the effectiveness of the proposed method, especially compared to the LLM and deployed to downstream tasks.

**Reasons To Reject:**

1. The writing, especially in the introduction, could be improved. It is difficult to fully understand the shortcomings of existing methods and the novelty of utilizing local and global information jointly.

**Reproducibility:**

4: Could mostly reproduce the results, but there may be some variation because of sample variance or minor variations in their interpretation of the protocol or method.

**Reviewer Confidence:**

2: Willing to defend my evaluation, but it is fairly likely that I missed some details, didn't understand some central points, or can't be sure about the novelty of the work.

---

> ### Author Rebuttal · Authors · 2023-08-29
>
> We express our gratitude to the Reviewer for providing us with valuable feedback. The following paragraphs address your reason-to-reject and question.
>
> # Shortcomings of Existing Methods
>
> Thank you for bringing up the issue with the writing in the introduction. We appreciate your feedback and will make sure to improve it in the camera-ready version, following your advice. Before we do that, we would like to further clarify the shortcomings of existing methods and highlight how our method stands out.
>
> Firstly, it is important to note that nodes in CSKGs often exist in non-canonicalized and free-form text. Despite having different descriptions, these nodes can still possess related semantics. However, existing learning-based methods such as TransE and RotatE are **unable to capture the semantic information of these nodes** (lines 49-71).
> Similarly, path-learning and neighbor-learning based methods also **fall short of capturing this semantic information** (lines 72-80). For instance, `brush your teeth` (line 72) is an isolated node in ConceptNet that cannot be distinguished based on any structural information.
> Furthermore, methods that rely on logical rules **suffer from sparsity issues and have limited generalizability** when applied to diverse rules in CSKGs (lines 81-100).
> In the case of CSKG completion tasks, existing methods **either require extensive fine-tuning or lack sensitivity to negative commonsense knowledge**, which is crucial for noise detection (lines 111-126).
>
> Overcoming all of these challenges, we propose a novel method that addresses each of these issues with different approaches. These proposed treatments form the novelty of our method, as explained in more detail below.
>
> # Novelty of Our Method
>
> Thanks for raising the issue of novelty. We would like to clarify that our work differs from a simple combination of two existing approaches, and its novelty primarily lies in two aspects: **(1) We introduce a new task: CSKG denoising, which can be applied to various CSKG construction and LLM distillation works. (2) We propose a novel approach that leverages both local and global information from textual data and graphs.**
>
> We would like to highlight that **we are the first to propose the denoising task upon Commonsense Knowledge Graphs (CSKGs) and associate that with an effective framework**. In Section 6, we show that models in our proposed task **bring significant improvements on downstream tasks** such as Commonsense Question Answering (CSQA): denoising models in our proposed task  (i.e., our proposed method to ATOMIC10X [1]) brings a 7.0% improvement in the PIQA dataset and a 6.4% improvement in the CSQA dataset when evaluating a zero-shot QA model trained on our denoised ATOMIC-10X. Those significant improvements demonstrate that **this task holds great importance in the present context, especially with the emergence of more distillation works from large language models**. Unlike the traditional approach of human annotation and critic filtering, our task provides a distinct option for cleaning distillation data. **Our denoising framework can be applied to any CSKG without the need for additional annotation or supervision signals, making it a performant yet easy-to-use plugin for CSKGs**. Thus, our proposed task and our accompanying method serve as a novel and impactful contribution to the NLP community.
>
> We would like to emphasize that our approach stands out by being the first to extract path-based logical rules and employ an RNN-based rule encoder to generalize them for denoising purposes. Figure 2 illustrates the joint training process, where a rule encoder in the global component compares triple and rule similarity. The resulting information propagates to the bottom-level triple encoder, enabling simultaneous parameter updates in the local component. **This aspect distinguishes our framework from a mere combination of existing modules**. Moreover, we enhance our method by incorporating semantic information derived from Large Language Models (LLMs), enabling better adaptation to commonsense knowledge scenarios. Thus, we firmly believe that our proposed approach contributes beyond simply combining two strategies, but rather organically adapting them to the task of noise detection and the domain of commonsense reasoning.
>
> # Reference
>
> [1] Peter West, Chandra Bhagavatula, Jack Hessel, Jena D. Hwang, Liwei Jiang, Ronan Le Bras, Ximing Lu, Sean Welleck, and Yejin Choi. 2022. Symbolic knowledge distillation: from general language models to commonsense models. In Proceedings of the 2022 Conference of the North American Chapter of the Association for Computational Linguistics: Human Language Technologies, NAACL 2022, Seattle, WA, United States, July 10-15, 2022, pages 4602– 4625. Association for Computational Linguistics.
>
> ***We genuinely appreciate your input and hope that our previous responses have adequately addressed your concerns and will help you improve your scores. If you have any further questions, please don't hesitate to ask. We are more than happy to provide additional clarification and assistance.***

---

### Meta-Review · Area_Chair_Eg7n · 2023-09-15

**Recommendation:** 3

**Metareview:**

The authors present a denoising framework for Commonsense Knowledge Graphs (CSKGs), which often contain spurious data. Leveraging semantic information, global rules, and local structures, this framework differentiates between authentic and noisy triples. Emphasizing link prediction across two datasets using Recall and AUC metrics, the evaluations show the value of this approach in enhancing data quality, which is further illustrated through zero-shot commonsense question-answering tasks.

This paper is technically sound and well-executed; however, the impact of the results is incremental and not particularly surprising to the community.

---

### Decision · Program_Chairs · 2023-10-07

**Decision:**

Accept-Findings

**Comment:**

The authors present a denoising framework for Commonsense Knowledge Graphs (CSKGs), which often contain spurious data. Leveraging semantic information, global rules, and local structures, this framework differentiates between authentic and noisy triples. Emphasizing link prediction across two datasets using Recall and AUC metrics, the evaluations show the value of this approach in enhancing data quality, which is further illustrated through zero-shot commonsense question-answering tasks.

This paper is technically sound and well-executed; however, the impact of the results is incremental and not particularly surprising to the community.